# Glycoprotein Non-Metastatic Melanoma Protein B Restricts PRRSV Replication by Inhibiting Autophagosome-Lysosome Fusion

**DOI:** 10.3390/v15040920

**Published:** 2023-04-05

**Authors:** Yunfei Xu, Mengjie Wang, Lin Zhang, Yu Pan, Wenli Zhang, Wenjie Ma, Hongyan Chen, Lijie Tang, Changyou Xia, Yue Wang

**Affiliations:** 1State Key Laboratory of Veterinary Biotechnology, Heilongjiang Provincial Key Laboratory of Laboratory Animal and Comparative Medicine, Harbin Veterinary Research Institute, Chinese Academy of Agricultural Sciences, Harbin 150069, China; 2College of Veterinary Medicine, Northeast Agricultural University, Harbin 150030, China; 3College of Veterinary Medicine, Southwest University, Chongqing 400715, China

**Keywords:** PRRSV, GPNMB, autophagy

## Abstract

Glycoprotein non-metastatic melanoma protein B (GPNMB) is a transmembrane protein enriched on the surface of some cells, including melanoma, glioblastoma, and macrophages. GPNMB has been reported to have multifaceted roles, such as facilitating cell–cell adhesion and migration, stimulating kinase signaling, and regulating inflammation. Porcine reproductive and respiratory syndrome virus (PRRSV) is the leading cause of severe economic loss in the swine industry worldwide. In this study, the role of GPNMB was investigated in porcine alveolar macrophages during PRRSV infection. We observed that GPNMB expression was markedly reduced in PRRSV-infected cells. The inhibition of GPNMB by specific small interfering RNA led to an enhancement in virus yields, and GPNMB overexpression decreased PRRSV replication. Further studies revealed that the overexpression of GPNMB could induce the accumulation of autophagosome through inhibiting autophagosome-lysosome fusion. Using a specific inhibitor, we confirmed that the inhibition of autophagosome-lysosome fusion significantly inhibited viral replication. Taken together, our data demonstrate that GPNMB inhibits PRRSV replication by inhibiting the autophagosome-lysosome fusion and provides a novel therapeutic target for virus infection.

## 1. Introduction

Porcine reproductive and respiratory syndrome (PRRS), caused by PRRS virus (PRRSV), is one of the most economically important diseases in pigs worldwide, characterized by severe reproductive failure in sows and respiratory distress in piglets and growing pigs [1]. PRRSV is an enveloped, positive-sense single-stranded RNA virus, classified in the family *Arteriviridae* of the order *Nidovirales* [2,3]. The PRRSV genome is approximately 15.4-kb and encodes at least 10 open reading frames (ORFs) [4]. There are two major PRRSV groups based on the genetic differences seen in the prototype strains Lelystad (referred to as European-like strain) and VR2332 (referred to as North American-like strain) [5,6]. PRRSV primarily infects porcine alveolar macrophages (PAMs), as well as the monocytic lineage of other tissues [7].

In eukaryocytes, autophagy is a widely existing conservative mechanism that can transport long-lived cytoplasmic proteins and damaged organelles to lysosomes to be degraded for the maintenance of cellular homeostasis [8,9]. Autophagy is not only necessary for cellular homeostasis but also contributes to innate and adaptive immunity against a wide variety of intracellular microbial pathogens, including bacteria, viruses, and protozoa. For example, during Sindbis virus (SINV) infection, autophagy receptor p62/SQSTM1 binds to a SINV capsid protein and targets the viral capsid to the autophagosome [10]. Picornaviruses, including poliovirus, restrict viral infection by initiating the autophagic degradation of the viral RNA genome [11]. However, growing evidence suggests that the viruses have evolved to evade autophagy or to exploit autophagy for their own life cycles, such as SARS-CoV-2, human immunodeficiency virus, influenza A virus, and coxsackievirus [12,13,14,15,16,17]. For example, SARS-CoV-2 inhibits autophagy activity by blocking the fusion of autophagosomes/amphisomes with lysosomes [12]. In addition, some studies demonstrated that coxsackievirus used accumulated autophagosomes for their replication [16,17]. A previous study showed that PRRSV infection induced incomplete autophagy, showing an increased accumulation of autophagosomes [18]. However, others have shown that PRRSV infection induced complete autophagy [19,20]. Currently, the relationship between PRRSV and autophagy is incompletely understood.

Glycoprotein non-metastatic melanoma protein B (GPNMB) is a type I transmembrane protein that is also known as Osteoactivin (OA), dendritic cell heparan sulfate proteoglycan integrin-dependent ligand (DC-HIL), and hematopoietic growth factor inducible neurokinin-1 type (HGFIN) [21,22]. GPNMB is widely expressed in many tissues. Because of its functions in immune system activation, cell proliferation, angiogenesis, tissue repair, and the invasion and metastasis of malignant tumors, GPNMB has been implicated in various physiological and pathological processes [23,24]. Recent studies have generated a more complex picture regarding the expression of GPNMB in various cancer progression, including lung cancer, ovarian cancer, stomach cancer, and breast cancer [25,26,27]. However, there have been no reports concerning the relationship between GPNMB and virus infection. In this study, we showed that PRRSV infection downregulated GPNMB expression in PAMs. Additionally, silencing GPNMB facilitated PRRSV replication, and the overexpression of GPNMB restricted PRRSV replication. Successively, we demonstrated for the first time that GPNMB induced the accumulation of autophagosome through the inhibition of autophagosome-lysosome fusion. Taken together, this study identified a novel function of GPNMB that can restrict PRRSV replication.

## 2. Materials and Methods

### 2.1. Cells and Viruses

Primary PAMs were obtained from 5-week-old specific-pathogen-free pigs. The immortalized PAM cell lines were a gift from Dr. Yan-dong Tang [28]. Both primary and immortalized PAMs were cultured in RPMI-1640 medium (Gibco, Waltham, MA, USA), supplemented with 10% fetal bovine serum (FBS, Gibco, Waltham, MA, USA). Marc-145 cells and HEK-293T cells were cultured in Dulbecco’s modified eagle medium (DMEM, Gibco, Waltham, MA, USA) supplemented with 8% FBS. The HP-PRRSV strain HuN4 (GenBank no. EF635006) was grown and titrated in Marc-145 cells, as described previously [29]. To obtain replication-incompetent PRRSV, 10 mL of virus suspension was dispensed to form a layer of fluid in an open cell culture dish and was irradicated under UV light for 5 h with gentle shaking at intervals.

### 2.2. RNA Interference Assay

siRNA duplexes were designed specifically to knock down the endogenous expression of GPNMB (Table 1). The GPNMB siRNAs and negative control siRNA at the concentration of 60 nM were transfected into target cells for 24 h using Lipofectamine™ RNAiMAX reagent (Invitrogen, Carlsbad, CA, USA), according to the manufacturer’s instructions. At 24 h posttransfection, the cells were infected with PRRSV, followed by further analysis.

### 2.3. Plasmids Construction and Transfection

The full-length porcine GPNMB protein-coding sequence was amplified from PAMs cDNA using the primers listed in Table 2 and then cloned into the eukaryotic expression vector pCAGGS (Addgene, Cambridge, MA, USA) and the bicistronic lentivirus vector pLVX-IRES-ZsGreen1 (Addgene, Cambridge, MA, USA), respectively. The nucleotide sequences of the plasmids expressing GPNMB were determined to ensure that the correct clones were used in this study. When the cells reached approximately 80% confluence, they were transfected with recombinant plasmids or an empty vector. The packaging and production of lentiviral vectors were performed as previously described [30]. When immortalized PAMs reached 80% confluence, cell monolayers were transduced with a recombinant lentivirus. At 24 h postincubation, the cells were harvested for further analysis.

### 2.4. Viral Infection and Drug Treatment

The monolayers of primary PAMs and immortalized PAMs were infected with PRRSV at a multiplicity of infection (MOI) of 0.1 for 1 h at 37 °C. Unattached viruses were removed, and the cells were washed three times with phosphate-buffered saline (PBS, pH 7.0). The cells were then cultured in a complete medium for various time points until the samples had been harvested. For the autophagy induction and inhibition experiments, immortalized PAMs were pretreated with various concentrations of rapamycin, bafilomycin A1 and chloroquine or an equal volume of DMSO (carrier control) for 24 h prior to viral infection. Virus infection was then performed in the presence of these reagents. At 24 h post-infection, the cells were collected for subsequent analysis.

### 2.5. Quantitative PCR(qPCR)

qPCR was carried out as previously described [31] by using the specific primers listed in Table 3. The quantitative reactions were set up in triplicate using SYBR premixed Ex Taq (Takara, Japan). Briefly, the relative quantification was calculated by the cycle threshold (ΔΔCT) method [32].

### 2.6. Western Blot

Western blot analysis was undertaken as described previously [33], with slight modifications. The samples were separated by SDS-PAGE under reducing conditions and blotted onto polyvinylidene difluoride membranes (Merck Millipore, Temecula, CA, USA). The membranes were incubated with the appropriate primary and secondary antibodies, washed, and visualized with an Odyssey infrared imaging system (LI-COR Biosciences, Lincoln, NE, USA). The mouse anti-Flag monoclonal antibody (mAb) was purchased from Abcam (Cambridge, UK). The rabbit anti-GPNMB polyclonal antibody and mouse anti-β-actin mAb were purchased from Sigma (Northbrook, IL, USA). The IRDye 680 conjugated goat anti-mouse IgG and the IRDye 800 conjugated goat anti-rabbit IgG were produced by Li-Cor Biosciences (Lincoln, NE, USA). The mouse anti-PRRSV nucleocapsid (N) protein mAb was produced and purified in our laboratory. 

### 2.7. Virus Titration

Virus titration was performed as previously described with a slight modification [34]. Median tissue culture infectious dose (TCID50) assays were performed on Marc-145 cells for PRRSV following the Reed–Muench method [35]. Briefly, the cell monolayers were inoculated with serial dilutions of each virus stock and incubated for 3 days prior to the observation of the presence of a cytopathic effect.

### 2.8. Confocal Microscopy and Immunofluorescence Analysis (IFA)

Confocal microscopy was performed as described previously [29] with slight modifications. After various treatments, the HEK-293T and immortalized PAMs cell monolayers were fixed with 4% paraformaldehyde for 30 min. After blocking, the cells were incubated with mouse anti-Flag mAb or rabbit anti-HA mAb (Abcam, Cambridge, UK) for 1 h. After washing, the cells were incubated with Alexa Fluor 488-conjugated goat anti-mouse IgG and Alexa Fluor 594-conjugated goat anti-rabbit IgG (Invitrogen, Carlsbad, CA, USA) for 1 h. Lastly, the cells were stained with 4,6-diamidino-2-phenylindole (DAPI) (Biosharp, Hefei, China) for 5 min and then examined with a ZEISS Confocal Laser Scanning Microscopy (ZEISS, Jena, Germany). 

An IFA assay was carried out according to the protocol of confocal microscopy with slight modification. Briefly, primary PAMs and immortalized PAMs were subjected to infection with PRRSV after treatments, as indicated. At 24 h post-inoculation, the cells were fixed and incubated with anti-PRRSV N protein mAb and then incubated with FITC-conjugated goat anti-mouse IgG or TRITC-conjugated goat anti-mouse IgG (Thermo Fisher Scientific, Waltham, USA). Finally, the cells were visualized under an inverted fluorescence microscope equipped with a camera (ZEISS, Jena, Germany). The percentage of virus-positive cells was calculated.

### 2.9. Statistical Analysis

All of the statistical data were expressed as mean ± standard deviation (SD) of three independent experiments and analyzed using Student’s *t*-test (two-tails). A *p*-value of <0.05 was considered statistically significant.

## 3. Results

### 3.1. PRRSV Infection Induces GPNMB Downregulation

To investigate the expression pattern of GPNMB during PRRSV infection, we first analyzed the mRNA and protein levels of the GPNMB in virus-infected primary PAMs. The qPCR results showed that the mRNA level of GPNMB was significantly downregulated in PRRSV-infected primary PAMs at different MOI and different time points (Figure 1A,C). Similarly, the protein level of GPNMB was also significantly downregulated in PRRSV-infected primary PAMs (Figure 1B,D). Of interest, we did not observe a decrease in GPNMB in UV-PRRSV-infected primary cells (Figure 1E,F), suggesting that PRRSV replication is required for the induction of decreased GPNMB in target cells. Collectively, our data indicate that PRRSV infection decreases GPNMB expression.

### 3.2. Knockdown of Endogenous GPNMB Expression Facilitates PRRSV Infection

To investigate the role of the GPNMB protein in PRRSV infection, we designed a siRNA that targeted GPNMB and transfected into the primary PAMs and immortalized PAMs. The expression of the GPNMB was determined by qPCR and Western blot. As shown in Figure 2A,B, the cells transfected with GPNMB-specific siRNA showed a marked reduction in the levels of GPNMB mRNA and protein. At 24 h post-siRNA transfection, primary PAMs were inoculated with PRRSV for an additional 24 h. We observed that all three specific siRNAs resulted in a significant increase in the level of PRRSV RNA and protein (Figure 2C). Since viral RNA and protein levels can be upregulated by all three siRNAs (siRNA-1, siRNA-2, and siRNA-3), and the efficiency of siRNA-2 was the highest; thus, siRNA-2 was used to knock down GPNMB in the following experiments. The results of the IFA analyses showed that the number of PRRSV-positive cells in the GPNMB knockdown groups was notably more than in the control group (Figure 2E). Likewise, the knockdown of the GPNMB by siRNA-2 led to a significant increase in viral titers compared with the negative control by TCID50 assay (Figure 2G). Consistent with the results in primary PAMs, GPNMB-specific siRNA significantly increased the levels of virus RNA, viral protein and virus titers in immortalized PAMs as well (Figure 2D,H). Additionally, the IFA analysis indicated that the number of PRRSV-positive cells was significantly increased in GPNMB knockdown groups than in the control group (Figure 2F). Taken together, the knockdown of endogenous GPNMB expression in target cells results in the increased replication of PRRSV, indicating that GPNMB plays an important role in the viral infection process.

### 3.3. Overexpression of GPNMB Inhibits PRRSV Infection

To further determine the effect of GPNMB on PRRSV infection, we generated a GPNMB-overexpressing stable cell line using lentivirus transduction in immortalized PAMs. All of the cells were visually green-fluorescence positive under a fluorescence microscope (Figure 3A). The Western blot results confirmed that GPNMB was successfully overexpressed in immortalized PAMs (Figure 3B), and the results of the CCK8 assay showed that GPNMB overexpression did not affect the proliferation of immortalized PAMs (Figure 3I). After infection with PRRSV, we observed that viral replication was decreased in GPNMB-overexpressing PAMs compared to the vector control (Figure 3C,D). Furthermore, we examined the dynamic changes of viral replication at different time points in GPNMB-overexpressing PAMs. As shown in Figure 3E,F, we observed that the levels of PRRSV RNA and protein were significantly decreased at 6 hpi and after, suggesting that GPNMB regulates PRRSV infection at the stage of viral biosynthesis after the virus enters the host cell. The TCID50 data showed that viral yields were lower in GPNMB-overexpressing PAMs than in vector control (Figure 3H). Additionally, the number of PRRSV-positive cells in the GPNMB-overexpressing group was notably less than the control group (Figure 3G). Taken together, these data demonstrate that GPNMB negatively regulated PRRSV infection.

### 3.4. GPNMB Induce Autophagosome Formation

Given the fact that GPNMB colocalized with autophagy-related proteins LC3 [36], we speculated that GPNMB might participate in the process of autophagy. When autophagy is induced, a series of conjugation reactions lead to the conversion of cytosolic microtubule-associated LC3 (LC3-I) to the lipidated form LC3 (LC3-II), and the amount of LC3-II is correlated with the number of autophagosomes [37]. The plasmid pCAGGS-GPNMB-Flag was transfected into HEK293T and immortalized PAMs, and the expression of both forms of LC3 protein was detected by Western blot. The results showed that the expression levels of LC3-II were upregulated (Figure 4A,C), and the LC3-II to LC3-I ratio was significantly increased in GPNMB overexpressing cells relative to controls (Figure 4B,D), suggesting the activation of autophagic activity. To further confirm that GPNMB can induce the formation of autophagosomes, we checked the result in HEK293T and immortalized PAMs by confocal microscopy. As shown in Figure 4E,F, the EGFP-LC3 puncta accumulated in GPNMB-overexpressed cells, suggesting that GPNMB induces the formation of autophagosomes. Taken together, these data suggest that GPNMB induces autophagosome formation in cells.

### 3.5. GPNMB Blocks Autophagosome Degradation

Since the accumulation of autophagic vacuoles can be indicative of two different processes: increasing autophagosome formation or a reduction in autophagosome degradation, we investigated the function of lysosomes, which are responsible for the digestion of autophagic substrates. To probe this notion, we monitored autophagy flux using the EGFP-RFP-LC3 reporter system [38,39]. As shown in Figure 5D, the Rapamycin control increased both yellow (autophagosomes) and red (autolysosomes) signals, indicating that the rapamycin control induces the complete autophagic process. However, the overexpression of GPNMB dramatically impaired the transition of EGFP-RFP-LC3-positive autophagosomes (yellow signals) to RFP-positive/EGFP-negative autolysosomes (red signals). To rule out the possibility that autophagosomes fused with lysosomes but were not efficiently acidified in the transfected cells, we investigated the colocalization of EGFP-LC3 with lysosomal-associated membrane protein 1 (LAMP1), a lysosome marker. The colocalization of EGFP-LC3 and RFP-LAMP1 was observed in cells treated with Rapamycin (Figure 5E). In contrast, we were not able to observe the colocalization of EGFP-LC3 and RFP-LAMP1 in cells transfected with GPNMB, suggesting the inhibition of fusion between autophagosomes and lysosomes. For further confirmation, we labeled lysosomes with red fluorescent Lysotracker to perform live cell imaging according to the manufacturer’s instructions. Immunofluorescence analysis showed that both autophagosome and functional lysosome were significantly increased in GPNMB overexpressed cells, indicating a decrease in the formation of autolysosomes (Figure 5F). In contrast, Rapamycin-treated cells, as the positive control of autophagy, showed the merged yellow signals of autophagosome and functional lysosome. In addition, Western blot results showed that the p62/SQSTM1 protein level was significantly increased in HEK293T and GPNMB-overexpressing PAMs (Figure 5A,B), and the p62/SQSTM1 protein level was significantly decreased in GPNMB-knockdown immortalized PAMs (Figure 5C). Taken together, the results above indicated that GPNMB induced the accumulation of autophagosomes by inhibiting autophagosome-lysosome fusion.

### 3.6. Bafilomycin A1 and Chloroquine Inhibit Virus Replication by Decreasing Autophagosome Degradation

To define the impact of incomplete autophagy on viral infectivity, we used two autophagy inhibitors: bafilomycin A1 and chloroquine. The results of cytotoxicity of bafilomycin A1 and chloroquine showed no significant cytotoxicity on immortalized PAMs (Figure 6I). The effect of bafilomycin A1 and chloroquine inhibition on the degradation of the late-stage autophagy pathway was first accessed, and the results showed that the protein levels of p62/SQSTM1 increased in cells treated with the bafilomycin A1 and chloroquine (Figure 6A,B). In the next, immortalized PAMs were pretreated with bafilomycin A1 or chloroquine and then infected with PRRSV. As shown in Figure 6C–F, viral protein and viral RNA levels were decreased in immortalized PAMs treated with bafilomycin A1 or chloroquine in time- and dose-dependent manners. Additionally, the IFA analysis indicated that the number of PRRSV-positive cells was significantly lower in bafilomycin A1 or chloroquine-treated immortalized PAMs than that in a mock-treated control (Figure 6G). The reduced titers of viruses in infected cell cultures containing bafilomycin A1 or chloroquine were further confirmed by measuring the TCID50 (Figure 6H). Taken together, these data suggest that the degradation of the late-stage autophagy pathway is important for PRRSV replication, and the inhibition of this process impairs PRRSV replication.

## 4. Discussion

GPNMB plays an important role in immune system activation and cell adhesion and migration functions in humans [23,24]. In this study, we investigated the antiviral role of host GPNMB in viral infection. We found that GPNMB overexpression strongly inhibited PRRSV replication, whereas GPNMB knockdown facilitated PRRSV replication. Moreover, we found that PRRSV infection reduced GPNMB expression. However, how GPNMB regulates viral replication will require further investigation. One study has confirmed that GPNMB colocalized with autophagy-related proteins LC3 [36]. Thus, we speculated that GPNMB might participate in the process of autophagy. Subsequently, we found that GPNMB induced the accumulation of autophagosomes. By different approaches, including non-cleavable p62/SQSTM1 degradation, EGFP-RFP-LC3 reporter, LAMP1/LC3 co-localization and LysoTracker/LC3 co-localization, we further demonstrate that GPNMB results in the significant inhibition of autophagic flux by limiting the fusion of autophagosomes with lysosomes. All these findings indicate that GPNMB can induce the accumulation of autophagosome by inhibiting autophagosome-lysosome fusion. However, whether GPNMB-induced accumulation of autophagosome affects PRRSV replication needs further investigation.

Recent studies have demonstrated that the infection processes of viruses are closely related to the autophagy of host cells. During infection, autophagic processes target intracellular viruses to autophagosomes for degradation. Intracellular pathogens have developed various molecular strategies to evade or subvert autophagy for their own benefit [40]. Because the fusion of autophagosomes with lysosomes is a main step in the autophagic flux, viruses might disrupt this process for their benefit. Several studies have demonstrated that many viruses can inhibit the fusion between autophagosomes and lysosomes, such as HIV-1, rotavirus, coxsackievirus B3, and human parainfluenza virus [41,42,43,44]. Therein, the phosphoprotein of the Human Parainfluenza Virus suppresses autophagosome-lysosome fusion and causes the accumulation of autophagosomes by interacting with SNAP29 to increase virus production [44]. Similarly, the Nsp4 protein of rotavirus binds to autophagosomes and inhibits their fusion with lysosomes to enhance viral RNA replication [42]. However, others have shown that L-asparagine inhibits fusion between autophagosomes and lysosomes, resulting in decreased DENV-3 production [45]. In addition, the depletion of either LAMP2 or Rab7, which allowed the accumulation of autophagosomes by preventing fusion with lysosomes, also inhibited HCV viral replication [46]. Interestingly, in our study, we found that autophagy inhibitor chloroquine and bafilomycin A1 significantly suppressed the PRRSV proliferation in immortalized PAMs. Previous studies showed that PRRSV infection induced complete autophagy [19,20]. However, others have shown that PRRSV infection induced incomplete autophagy shows an increased accumulation of autophagosomes due to the inhibition of autophagosome-lysosome fusion [18]. Autophagy plays different roles in different kinds of viral infections. To date, the role of autophagy in virus replication has been controversial. 

Positive-stranded RNA (+RNA) viruses that belong to the *Nidovirales*, such as MHV and equine arteritis virus (EAV), utilize virus-induced double-membrane vesicles (DMVs) as membrane scaffolds for replication and transcription [47,48,49,50,51]. As *Arterivirus* EAV, PRRSV also appropriates DMVs into replication ‘factories’ [52]. The viral RNA products are localized in the DMVs lumen and transported to the cytosol for translation and virion assembly via double-membrane-spanning molecular pores. The latest studies have found that MHV Nsp3 is the main component of the double-membrane-spanning molecular pores located in the MHV-induced DMVs membrane [53]. DMVs formed in virus-infected cells are smaller than regular autophagosomes. They exhibit hallmarks of autophagosomes/amphisomes, such as positivity for LC3 and the late endosomal/lysosomal marker LAMP1, but their further maturation into degradative autolysosomes is blocked [43,54,55,56]. Although DMVs exhibit hallmarks of autophagosomes, the data described above indicate that DMVs may arise from the viral hijacking of the host autophagy pathway, and there is evidence for the involvement of autophagy proteins in DMVs generation in rhinovirus- and poliovirus-infected cells [57]. Similar to the generation of MHV-induced DMVs, during PRRSV infection, autophagosome may also be modified by PRRSV nonstructural proteins and subsequently generate PRRSV-induced DMVs. In this study, we found that GPNMB induced the accumulation of autophagosomes through inhibiting autophagosome-lysosome fusion. The accumulation of autophagosomes may suppress PRRSV-induced DMVs formation through competitive inhibition, thereby inhibiting viral RNA replication. 

In summary, we demonstrate that host protein GPNMB can suppress PRRSV replication by inhibiting autophagosome-lysosome fusion. Although the exact mechanisms by that GPNMB affects autophagosome-lysosome fusion remain to be further elucidated, we have demonstrated that PRRSV infection results in the downregulation of GPNMB and that GPNMB plays an important role in the replication stage of PRRSV. In this study, we have uncovered a novel mechanism for understanding the roles of host protein GPNMB in PRRSV replication.

## Figures and Tables

**Figure 1 viruses-15-00920-f001:**
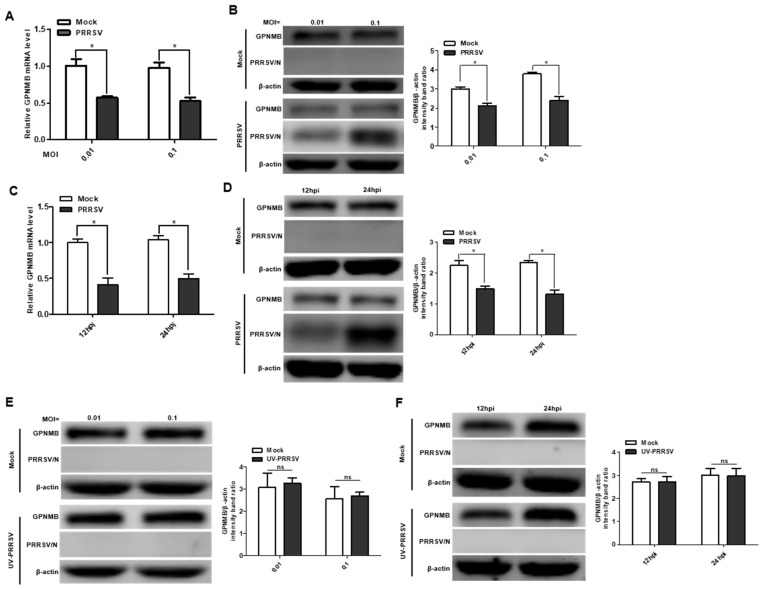
PRRSV infection alters GPNMB expression in cells. (**A**–**D**) PRRSV infection leads to the downregulation of GPNMB. Primary PAMs were infected with PRRSV at different MOI, and the samples were obtained at 24 h. The levels of GPNMB RNA and protein were determined by qPCR and Western blot (**A**,**B**). Primary PAMs were infected with PRRSV at an MOI of 0.1, and samples were obtained at different time points. The levels of GPNMB RNA and protein were determined by qPCR and Western blot (**C**,**D**). (**E**,**F**) The UV-PRRV incubation does not affect the GPNMB protein level. Primary PAMs were incubated with UV-PRRSV, and the cells were then further cultured for the indicated times. The cell lysates were subjected to immunoblotting with the antibodies indicated. Results are representative of three independent experiments (means ± SD). ns (no significant difference) *p* > 0.05; * *p* < 0.05. The *p*-value was calculated using Student’s *t*-test.

**Figure 2 viruses-15-00920-f002:**
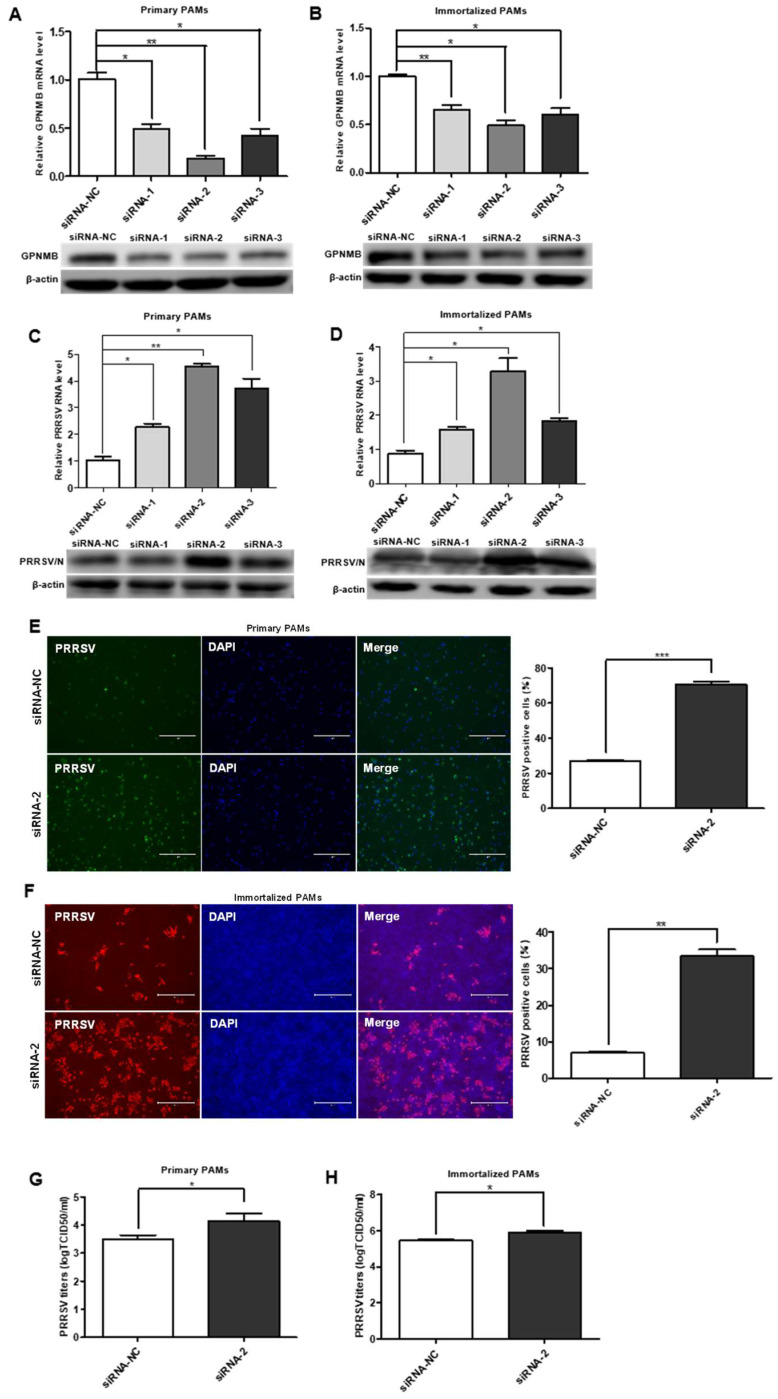
Knockdown of the endogenous GPNMB expression increases PRRSV infection. (**A**,**B**) GPNMB siRNA reduced endogenous GPNMB expression. Primary PAMs (**A**) and immortalized PAMs (**B**) were transfected with siRNA-GPNMB or siRNA-NC for 24 h, and the knockdown efficiency of GPNMB was determined by quantitative qPCR and Western blot. (**C**–**H**) Knockdown endogenous GPNMB expression facilitates PRRSV infection. Cells were transfected with siRNA-GPNMB or siRNA-NC for 24 h, and cells were then exposed to PRRSV for 24 h. The levels of viral RNA and protein in Primary PAMs (**C**) and immortalized PAMs (**D**) were determined by quantitative qPCR and Western blot; Primary PAMs (**E**) and immortalized PAMs (**F**) cell monolayers were fixed and examined for virus infection by IFA; and virus titers in Primary PAMs (**G**) and immortalized PAMs (H) were evaluated with TCID50. Results are representative of three independent experiments (means ± SD). * *p* < 0.05; ** *p* < 0.01; *** *p* < 0.001. The *p*-value was calculated using Student’s *t*-test.

**Figure 3 viruses-15-00920-f003:**
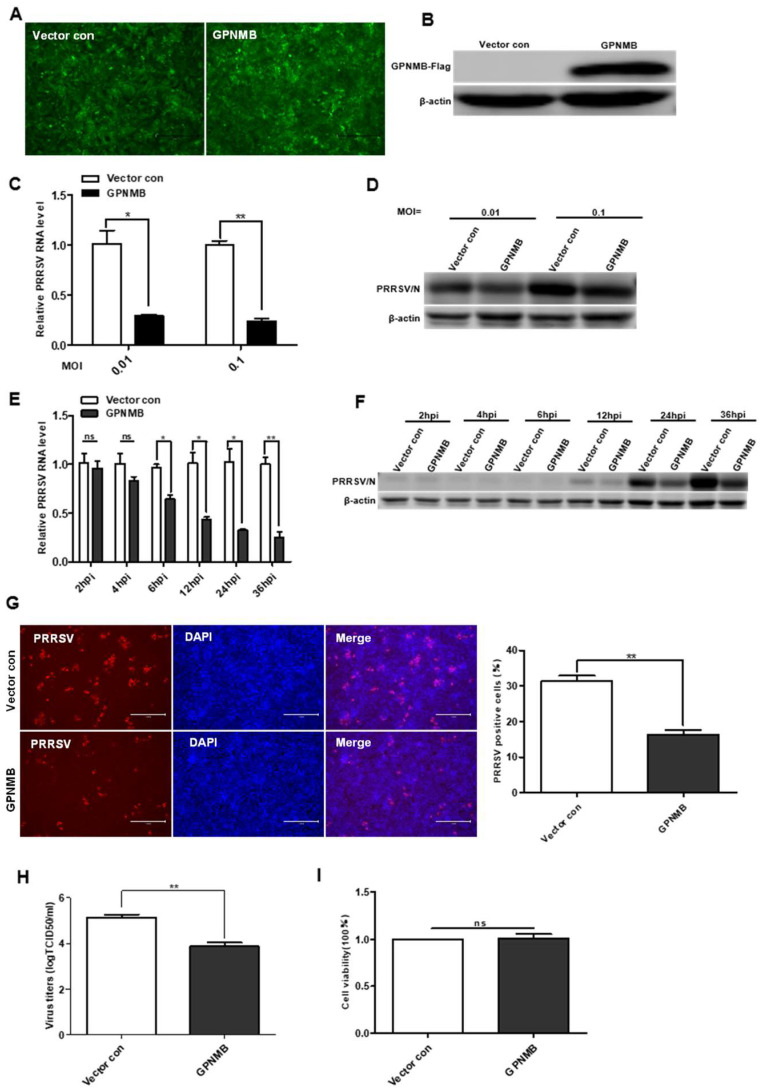
Overexpression of GPNMB inhibits PRRSV infection. (**A**) Immortalized PAMs were transduced with a bicistronic lentivirus vector expressing ZsGreen alone (vector control) or GPNMB plus ZsGreen as described in Materials and Methods. (**B**) Western blot analyses detected the protein level of GPNMB in cells infected with the GPNMB overexpression lentivirus. (**C**–**H**) GPNMB overexpression inhibits PRRSV infection. The GPNMB-overexpressing PAMs and the vector control were infected with the PRRSV at different MOI, respectively. The levels of viral RNA and viral protein were determined by qPCR and Western blot (**C**,**D**); The GPNMB-overexpressing PAMs and the vector control were exposed to PRRSV at an MOI of 0.1, and samples were obtained at different time points. The levels of viral RNA and viral protein were determined by qPCR and Western blot (**E**,**F**); Cells were infected with PRRSV at an MOI of 0.1 for 24 h. Cell monolayers were fixed and examined for virus infection by IFA (**G**), and virus titers were evaluated with TCID50 (**H**). (**I**) CCK8 assay assessed the proliferation ability of GPNMB-overexpressed PAMs. Results are representative of three independent experiments (means ± SD). ns *p* > 0.05; * *p* < 0.05; ** *p* < 0.01. The *p*-value was calculated using Student’s *t*-test.

**Figure 4 viruses-15-00920-f004:**
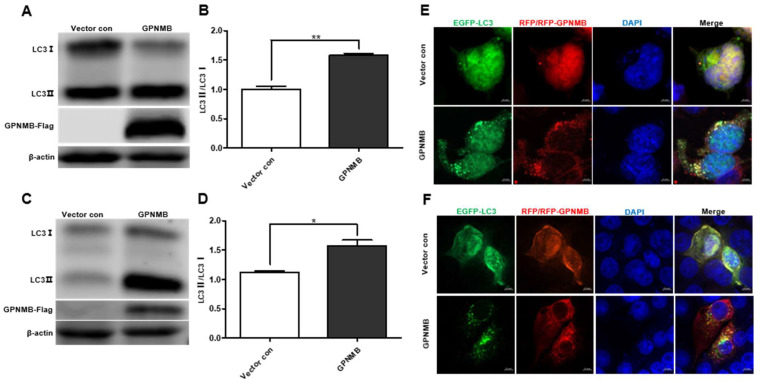
The formation of autophagosomes is induced by GPNMB. (**A**–**D**) LC3 protein conversion. HEK293T cells (**A**,**B**) and immortalized PAMs (**C**,**D**) were transfected with GPNMB-Flag; and then harvested and lysed. Extracts were analyzed by Western blot using an anti-LC3 antibody. The ratio of LC3II/LC3I reflects the level of autophagy. (**E**,**F**) EGFP-LC3 and RFP-GPNMB were co-transfected into HEK293T (**E**) and immortalized PAMs (**F**). Twenty-four hours after transfection, cells were harvested and visualized under a fluorescence microscope. Nuclei were stained with DAPI. * *p* < 0.05; ** *p* < 0.01. The *p*-value was calculated using Student’s *t*-test.

**Figure 5 viruses-15-00920-f005:**
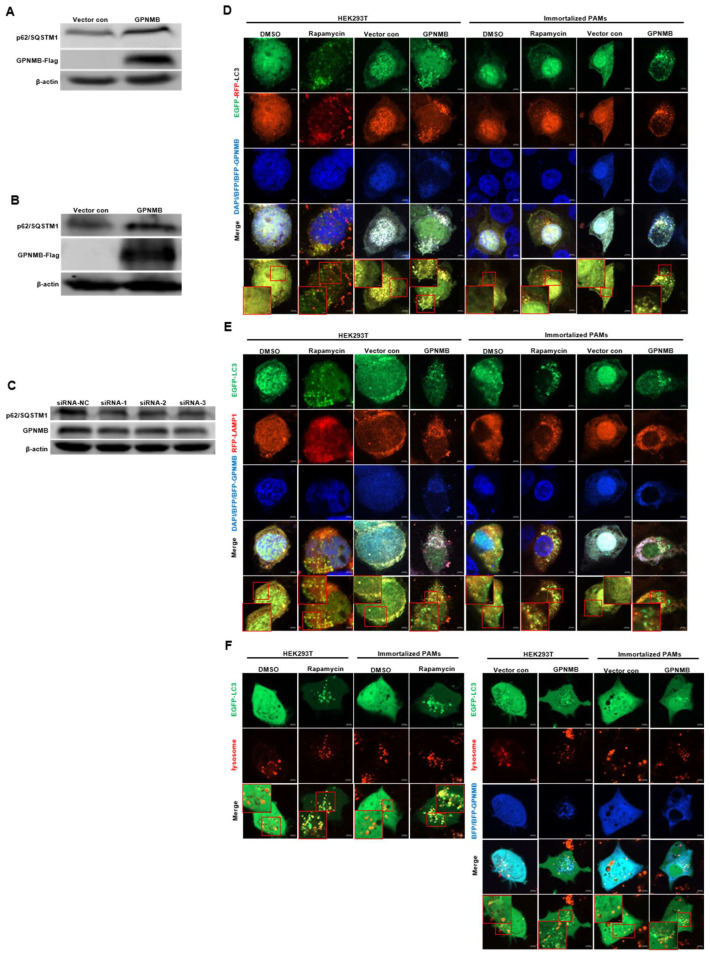
Autophagosomes do not fuse with lysosomes in GPNMB-overexpressed cells efficiently. (**A**,**B**) HEK293T cells (**A**) and immortalized PAMs (**B**) were transfected with GPNMB-Flag and then harvested and lysed. Extracts were analyzed by Western blot using an anti-p62/SQSTM1 antibody. (**C**) Immortalized PAMs were transfected with siRNA-GPNMB or siRNA-NC for 24 h, and then harvested and lysed. Extracts were analyzed by Western blot using an anti-p62/SQSTM1 antibody. (**D**) The colocalization analysis of HEK293T cells and immortalized PAMs overexpressed with BFP-GPNMB after transfected EGFP-RFP-LC3, treatment with rapamycin induced autophagy as a positive control. (**E**) The colocalization analysis of EGFP-LC3 with RFP-LAMP1 in GPNMB-overexpressed HEK293T cells and immortalized PAMs, treatment with rapamycin induced autophagy as a positive control. (**F**) Live cell imaging was performed to analyze colocalization of LysoTracker-stained acidified vesicles and EGFP-LC3-positive autophagosomes in GPNMB-overexpressed HEK293T cells and immortalized PAMs, treatment with rapamycin induced autophagy as a positive control. The red box indicates the regions where image was magnified.

**Figure 6 viruses-15-00920-f006:**
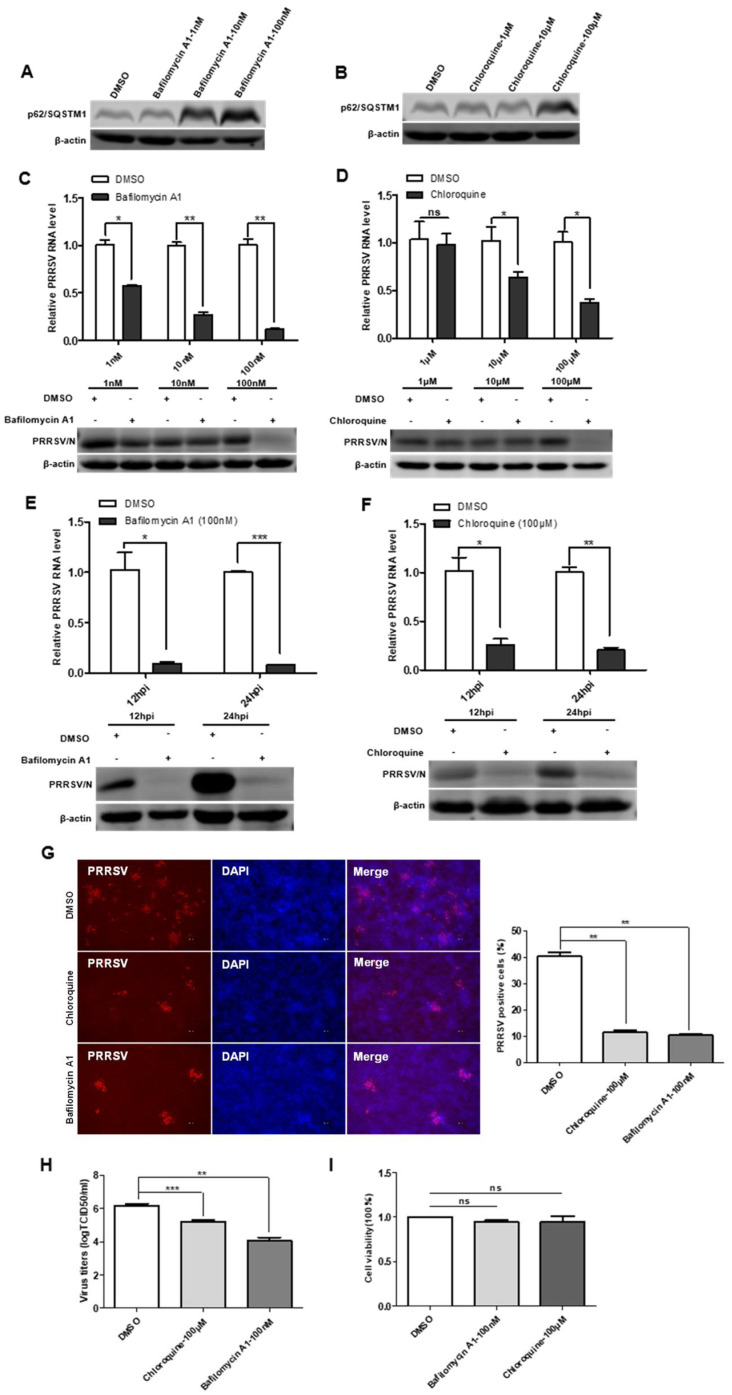
Autophagy inhibitors reduce PRRSV infection. (**A**,**B**) Bafilomycin A1 and chloroquine inhibit autophagosome degradation. Immortalized PAMs were pretreated with DMSO or inhibitors, Bafilomycin A1 and Chloroquine, at different concentrations for 24 h, and then harvested and lysed. Extracts were analyzed by Western blot using an anti-p62/SQSTM1 antibody. (**C**–**F**) Reduction of PRRSV RNA and N protein levels by autophagy inhibitors is dose- and time-dependent. Viral RNA in immortalized PAMs was determined by qPCR; The levels of viral N protein levels in immortalized PAMs were analyzed by Western blot. (**G**) Autophagy inhibitors decrease the number of PRRSV-positive cells. Immortalized PAMs were pretreated with DMSO or autophagy inhibitors, Bafilomycin A1 and Chloroquine, at different concentrations for 24 h. After washing, the cells were infected with PRRSV or mock control in the absence or presence of inhibitors. At 48 hpi, the cell monolayers were fixed and examined for PRRSV infection by IFA with an anti-PRRSV N protein mAb. The number of PRRSV-positive cells was calculated. (**H**) Virus titers were reduced after inhibitor treatment, as detected by the TCID50 assay. (**I**) Effects of inhibitors on cell proliferation. Immortalized PAMs were treated with the carrier control DMSO or autophagy inhibitors, Bafilomycin A1 and Chloroquine, at the indicated concentrations for 24 h. Cell cytotoxicity was analyzed with the CCK-8 system as described in Materials and Methods. Results are representative of three independent experiments (means ± SD). ns *p* > 0.05; * *p* < 0.05; ** *p* < 0.01; *** *p* < 0.001. The *p*-value was calculated using Student’s *t*-test.

**Table 1 viruses-15-00920-t001:** Sequences of sense strand of siRNA against the target gene in PAM.

RNA Oligo Name	Sequence (Positive Strand) (5′–3′)
Negative Control	UUCUCCGAACGUGUCACGUTT
siRNA-1	CCAGCCAAGGCCAUCACAATT
siRNA-2	CCACACACUUGGUCAGUAUTT
siRNA-3	CCAUACCUAUGUGCUCAAUTT

**Table 2 viruses-15-00920-t002:** Primers used for plasmid construction.

Primer Name	Primer Sequence (5′–3′)
pCAGGS-GPNMB-CDS-F-KpnI	CGGGGTACCATGGAATGTCTCTACTGTTTTCT
pCAGGS-GPNMB-CDS-R-XhoI	CCGCTCGAGGTTCTTGAGCAGTGGATCTTTCTCC
pLVX-GPNMB-CDS-F-XhoI	CCGCTCGAGATGGAATGTCTCTACTGTTTTCT
pLVX-GPNMB-CDS-R-NotI	ATTGCGGCCGCGTTCTTGAGCAGTGGATCTTTCTCC

**Table 3 viruses-15-00920-t003:** Primers used for relative quantitative RT-PCR.

Primer Name	Primer Sequence (5′–3′)
PRRSV-ORF7-F	AGATCATCGCCCAACAAAAC
PRRSV-ORF7-R	GACACAATTGCCGCTCACTA
Porcine-β-actin-F	CTTCCTGGGCATGGAGTCC
Porcine-β-actin-R	GGCGCGATGATCTTGATCTTC
Porcine-GPNMB-F	CAGGGGAGCATCCCCACGGA
Porcine-GPNMB-R	AAGGGTGCTCGTGAGGGCCA

## Data Availability

The data that support the findings of this study are available from the corresponding author upon reasonable request.

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
