# Peer review of "Glycoprotein Non-Metastatic Melanoma Protein B Restricts PRRSV Replication by Inhibiting Autophagosome-Lysosome Fusion"

_viruses, 2023, doi:10.3390/v15040920_

Round 1
Reviewer 1 Report
In this manuscript, the authors studied the role of host protein GPNMB in PRRSV infection using a series of techniques. They concluded that GPNMB can suppress PRRSV replication by inhibiting autophagosome-lysosome fusion. Overall, the scientific proof is strong. The manuscript is well-written and easy to follow. Please find detailed comments below.
Introduction:
The authors have provided detailed information on PRRSV, autophagy, and GPNMB individually. However, it is not clear how these three components are related to each other. The rationale for investigating the role of GPNMB in PRRSV infection is not well established. Furthermore, the relationship between GPNMB and autophagy remains unclear. It would be beneficial for the authors to provide a more comprehensive explanation of the underlying mechanism linking PRRSV, autophagy, and GPNMB.
Results:
Figure 1F, the expression of GPNMB in mock and UV-PRRSV group showed a similar pattern with a higher expression level at 24hpi than 12hpi. What is the explanation for this? Moreover, other than showing western-blot pictures, would be nice to include a quantitative analysis as well.
Figure 2, it is necessary to already indicate in the figures the results from primary PAMs and the ones from immortalized PAMs. Moreover, the quality of immunofluorescent pictures must be improved, especially the ones for primary PAMs where even the nucleus is not clearly visible.
Figure 3I, is it also data of three experiments? The error bar is not visible.
For all the figures with relative mRNA level, the controls were all set to 1 without error bars. This isn’t correct. Please make an average of data from three experiments, and then calculate the relative values of each data to this average. Please present the controls in this way so that the variation among three experiments could be presented.
Some minor comments:
Line 22, please change “led to a enhance in…” to “led to an enhancement in…”
Line 43-44, please change “to degrade” to “to be degraded”
Line 48-49, please correct the grammatical mistake
Line 52 & 53, please remove “COVID-19 virus”
Line 55, please change “use” to “used”
Line 57, please change “shown as increased accumulation…” to “shown as an increased accumulation…”
Line 86, please remove “,” and change “irradicated with UV light for 5 h by gentle shaking…” to “irradicated under UV light for 5 h with gentle shaking…”
Line 94, please specify what is “the indicated analysis”
Line 97, please change “Porcine GPNMB protein entire coding sequence…” to “The full-length of porcine GPNMB protein-coding sequence…”
Line 100, please remove “, and sequenced”. This sentence is long enough.
Line 137, please change “The virus titer was performed…” to “Virus titration was performed…”
Line 359, Please change “Study has…” to “One study has…”
Line 365, please remove the comma between “… significant, inhibition of …”
Line 366 & Line 384, please add “the” before “accumulation”
Line 387, please remove the possible extra space before “Previous…”
Reviewer 2 Report
Author here in this manuscript firstly stated GPNMB protein can inhibit PRRSV replication based on two observations: PRRSV protein has reduction by overexpression GPNMB and PRRSV replication is better in the PAM cell knocked down of GPNMP protein via specific siRNA. This inhibition is explained by inhibiting the autophagosome-lysosome fusion. It appreciated author has started idea and designed experiments in a logical way. However, what data provided in this manuscript didn’t support well, or not clean enough even for first conclusion, therefore, I think this manuscript is not well prepared yet for publishing.
Here are a few points related to first conclusion for instance, which especially the bad resolution of immunofluorescence and no proper negative control are not acceptable. Therefore, I will not continue to review the rest of manuscript.
1. Figure 1: author showed mRNA level of GPNMP has reduced due to PRRSV infection. The mRNA level has roughly 50% of reduction at infection of 12h and 24 hours in figure 1C, why the reduction is more significant with “two stars” in figure 1A, even less 50% of reduction at moi 0.01 infection? Also the author should state how many hours were samples collected in infection of moi 0.01 and 0.1. Has author looked the late time points when collecting sample after infection? This should be showed or stated at least.
2. Figure 1B and 1D showed protein level of GPNMP has reduced due to PRRSV infection. Proteins normally should load or run in one BLOT in order to compare the intensity of difference, otherwise, it is not comparable due to vary of exposure. Here the way author presented or organized is not easily understandable, it is confused whether those samples from same BLOT? Did author use two Mocks, one for infection at moi 0.1, other for moi 0.01? Where is BLOT for PRRSV/N in mock as control? Otherwise, no way to tell the PRRSV/N band observed is specific. Again, the reduction of GPNMP protein level is large under infection, even GPNMP band was completely absent at 12 hour in figure 1D, so why reduction is not significant at infection of 0.01 and 0.1 moi in figure 1B? What made this difference?
3. Figure 1E and 1F, author want to show the UV deactivated PRRSV has no effect on expression of GPNMP, the question is how author to convince the audience PRRSV no replication there, without showing BLOT of N of PRRSV in the control?
4. Figure 2, the resolution of immunofluorescence is quite low, even hard to tell where the nucleus staining by DAPI. The GPNMP knocked down by siRNA2 in both PAM and immortalized PAM has significant effect on PRRSV infection, which has increased 30-50% infected cells in figure E and F, thus it expects to see such effect on virus titer too, However, it is not case in figure 1G and H. Author may try to look at virus titers in different time points?
